# OpenReview forum: "Probabilistic Uncertain Reward Model"
_ICLR.cc/2026/Conference — Submitted to ICLR 2026_

### Official Review · Reviewer_y6mz · 2025-10-29

**Soundness:** 3
**Presentation:** 3
**Contribution:** 3
**Rating:** 4
**Confidence:** 4

**Summary:**

This paper introduces the Probabilistic Uncertain Reward Model (PURM), a generalization of the
classical Bradley–Terry Reward Model (BTRM) that represents rewards not as scalars but as Gaussian distributions.
The authors derive the corresponding loss function for PURM and propose a new metric to quantify uncertainty.
Empirically, they show that PURM matches existing reward models performance when predicting the reward, pro-
vides a sometimes more reliable measure of uncertainty, and seems to help mitigate reward hacking when used to train an LLM policy
while improving win rates.

**Strengths:**

1. The challenge of addressing both uncertainty and reward hacking in reward models is interesting, and this paper makes a step in this direction. Modeling the reward as a normal variable seems novel.
2. Proofs are written step by step and are easy to follow.

**Weaknesses:**

1. Conclusions about hyperparameter choice seems to be drawn from experiments by running a single seed, which
is far from ideal given the relatively small differences shown in the figures. This makes the results far less convincing. In particular, there seems not to be a large (or any?) difference between using the overlap-measure of uncertainty or just simply the sigma (empirically, that is).
2. Some metrics, such as length-controlled win rate, are not clearly defined or explained.
3. Some inconsistencies between loss in the equation and the one coded up, see below in questions.

**Questions:**

1. What is reason for changing the judge in section 3.3 between Figures 5(a) and 5(b) ?
2. In your implementation of the PURM loss in the code and in Appendix B, you used the average of log sigmoid(z),
whereas the straightforward loss from equation 4 is using the log of average of sigmoid(z). This seems to be a discrepancy that you do not mention. The loss you seem to use is an upper bound via Jensen’s inequality, can you explain why not simply use the log of average of sigmoid(z) instead ?
3. In section 3.2 you claim that your measure of uncertainty performs better, but figure 5(a) shows that using
just sigma gives comparable performance (given the experiment uses only one seed). How does the average of sigma
or the average of sigma divided by mu behave ? Can you make a convincing case that your measure performs better?
4. In figure 10, why do the curves with w=1e5 and w=1e6 differ even at early optimization steps < 400 when
there isn’t supposed to be any difference as the list size still hasn’t reached 1e5 ? And if this difference comes
from randomness during optimization then how and why did you conclude that 1e6 is the best hyperparameter
choice ?

[Extra question out of curiosity] DPO rewrites the BT reward to arrive at a loss that gets rid of the reward model alltogether. Can you do something similar for your case, when you assume the rewards are Gaussian normals?

---

> ### Author Response · Authors · 2025-11-15
>
> ## Thank you very much for reviewing our paper. Below are our responses to your comments.
>
> ### Regarding strengths:
> + Thank you very much for recognizing the motivation and theoretical proof of this paper!
>
> ### Regarding weaknesses:
> + W1: Thank you very much for the reminder and suggestion regarding the limitations of a single random seed. We have started to perform multi-seed repetitions of the core experiments. Due to the large computational cost of reinforcement learning experiments, these repeated runs require some time; we will complete them as soon as possible and submit the results at a later date. Thank you again for your constructive comments.
> + W2: We are very sorry for the confusion caused to your understanding. The length-controlled win rate is a metric adopted in Alpaca-Eval that is more robust to length preference relative to the win rate; therefore, we follow the official setting and directly adopt this metric. Thank you for the suggestion; we will add a detailed description of the metric in the experimental setup section in the revised version to more clearly describe the experiment settings.
>
> ### Regarding questions:
> + Q1: This is because we first designed experiments on the HH-RLHF dataset and, after observing that PURM showed the expected effects, we extended the experiments to Alpaca-Eval. For the Alpaca-Eval benchmark, we fully follow the official evaluation setup, and thus chose another model, GPT-4 Preview, as the judge.
> + Q2: Your question is very valuable! We are indeed optimizing a Jensen-inequality-based lower bound of the original likelihood. First, the Monte Carlo estimate of E[log σ(z)] is unbiased, and logsigmoid has a stable implementation (avoiding underflow when σ(z) approaches 0), making training more robust. Second, ∇θ log E[σ] = (1 / E[σ]) E[∇θ σ]; when E[σ] is very small or very large, the denominator leads to huge gradient magnitudes and high variance. In contrast, ∇θ E[log σ] = E[∇θ log σ], where ∂ log σ / ∂z = 1 − σ(z), with range in [0, 1], which is significantly more stable. Thank you again for your question. Due to space limitations, we did not include the above explanations in the previous version. We will carefully articulate the optimization techniques in the code implementation in the revised version to help readers better understand PURM’s lightweight implementation.
> + Q3: Thank you for the suggestion. For the random seed issue, we will supplement experiments as per the response in W1. Regarding the other two baselines you proposed, we are very sorry that we did not fully understand them: PURM first computes μ(x, y) and σ(x, y) for each sample (x, y), and then we consider two uncertainty measures, namely using σ(x, y) directly as u(x, y), or defining u(x, y) via overlap + marginalization as in Section 2.3 of the paper. How exactly are the “average of sigma or the average of sigma divided by mu” computed? Could you please provide further guidance? Thank you very much!
> + Q4: Thank you very much for your question! The differences before 400 steps are indeed caused by randomness. However, through ablation experiments on the window size w, we mainly focus on the overall trend of the RL performance curve as w increases. We can see that as w gradually grows, PURM becomes increasingly able to achieve better RL training results. We did not conclude that 1e6 is the best hyperparameter choice; rather, based on the ablation results, we believe the window size should be taken as large as possible to prevent the model from quickly forgetting early reward distributions, and to produce sufficiently reasonable uncertainty estimates for some later rewards. We will add the above analysis in the revised version according to your suggestion to better articulate the conclusions of the ablation experiments.
>
> ### Summary:
> + Again, thank you for your thoughtful and profound suggestions; they greatly help improve the theoretical soundness and experimental completeness of this paper. We would be very happy to continue the discussion with you!

---

> > ### Comment · Reviewer_y6mz · 2025-11-16
> > **Quick clarification on Q3**
> >
> > I thank the authors for the very swift first response which I will look at more deeply.
> > Just to clarify what I meant by "average sigma or average sigma/mu" for Fig 5a). It was to substitute your BC from Eq (6) simply with the mean of \sigma_1 and \sigma_2, or the mean of \sigma_1/\mu_1 and \sigma_2/mu_2. While they might be less principled, I was wondering how they fare as they strike me as simpler. Does that clarify?

---

> > > ### Author Response · Authors · 2025-11-17
> > >
> > > Thank you very much for your clarification. We understand the potential comparison methods you are concerned about, and we will promptly conduct the additional experiments and update you with the results at a later date.

---

> > ### Author Response · Authors · 2025-11-30
> >
> > Additionally, the RL training performance of PURM across multiple random seeds is now presented in Figure 8 of the revised manuscript. We appreciate your suggestion.

---

> ### Author Response · Authors · 2025-11-24
> **Experimental Results on Q3**
>
> Thank you again for your suggestions. We have uploaded a revised version of the paper through the Rebuttal Revision. In Figure 8 of Appendix 3.1, we present PPO training results that use either the mean of sigma_1 and sigma_2, or the mean of sigma_1/mu_1 and sigma_2/mu_2, as the BC coefficient. As shown, these variants perform worse than the RL training guided by the uncertainty estimates derived from the original BC formulation introduced in the main text. Thank you again for your suggestions! They further strengthen the comparisons and the persuasiveness of the method presented in our paper.

---

> > ### Comment · Reviewer_y6mz · 2025-11-27
> > **Thanks, no more questions**
> >
> > I thank the authors for your thorough response and following up on all my questions. I am satisfied with the responses and will raise my score.

---

### Official Review · Reviewer_CaMJ · 2025-10-31

**Soundness:** 2
**Presentation:** 3
**Contribution:** 2
**Rating:** 4
**Confidence:** 4

**Summary:**

This paper proposes the Probabilistic Uncertain Reward Model (PURM) as an extension of the classic Bradley–Terry reward model. Instead of producing a deterministic scalar reward, PURM outputs the parameters of a Gaussian distribution (μ,σ). The Bhattacharyya coefficient is further employed to measure the overlap between reward distributions, which serves as an uncertainty quantification mechanism. The authors claim that this approach improves the stability of RLHF training and mitigates reward hacking.

**Strengths:**

1. Clear motivation: The deterministic outputs of BTRM indeed lead to overconfidence and reward hacking risks. The paper addresses a practically relevant issue.

2. Simple and intuitive method: Modeling rewards as distributions and incorporating overlap-based uncertainty is straightforward and low-cost to implement.
3. Interesting uncertainty quantification: Using the Bhattacharyya coefficient, rather than variance alone, better aligns with the intuition of distributional separability.
4. Reasonable experimental coverage: Includes tests on public preference datasets and RLHF settings, along with ablation studies.
5. Practical applicability: Minimal code changes are required to integrate PURM into existing RLHF frameworks.

**Weaknesses:**

1. Limited novelty: Distributional reward modeling and uncertainty quantification are not new ideas. For example, URM[1] already proposed modeling reward uncertainty via probabilistic distributions. PURM is conceptually similar but does not clearly articulate its unique theoretical or empirical contributions.
2. Strong Gaussian assumption: Assuming reward distributions follow a Gaussian lacks justification. Real-world preference data may be skewed or multi-modal, raising concerns about robustness.
3. High sensitivity to λ: The effectiveness of the method strongly depends on the penalty coefficient λ, yet the paper provides no principled guidance or adaptive mechanism for its selection, limiting practical usability.

[1] Lou, Xingzhou, et al. "Uncertainty-aware reward model: Teaching reward models to know what is unknown." arXiv preprint arXiv:2410.00847 (2024).

**Questions:**

See weaknesses.

---

> ### Author Response · Authors · 2025-11-15
>
> ## Thank you very much for reviewing our paper. Below are our responses to your comments.
>
> ### Regarding strengths:
> + Thank you very much for recognizing the continuity and intuitiveness of the motivation and the ideas across sections, as well as the rationality of the experimental setup!
>
> ### Regarding weaknesses:
> + W1: Thank you very much for the detailed question! Our method, like [1], adopts probabilistic reward modeling, which is straightforward and easy to conceive, but [1] needs to introduce human-annotated reward values for maximum likelihood and regression training (line 454). However, human absolute rewards are often not sufficiently robust; a better approach is to train the reward model using human-annotated preference data ([2, 3]). Therefore, unlike [1], our method PURM does not need to introduce human-annotated absolute reward values and can be trained directly with existing conventional preference data, allowing reward estimation and uncertainty estimation to emerge spontaneously in this process. The entire process both has theoretical guarantees (Appendix A.1), can be implemented in a very lightweight manner (Appendix B), and runs with very little additional computational cost (Appendix C.4). We consider this to be the core contribution of this paper. Thank you again for the suggestion; we will expand the related work section in the revised version to more carefully explain the innovations of this paper relative to existing work.
> + W2: This is a very good question! We have also thought carefully about this, and the choice of the Gaussian distribution as the distribution modeled by PURM mainly includes the following two reasons:
>     - The Gaussian distribution is a commonly used assumption of unknown distributions in the field of machine learning: for example, in probabilistic principal component analysis and factor analysis [4] and in variational autoencoders [5], the Gaussian distribution assumption is used.
>     - To enable PURM to have the capability of “modeling arbitrary distributions,” we did consider adding a normalizing flow layer on top of the current Gaussian reward distribution of PURM, mapping to complex target distributions through invertible transformations. This approach is, in theory, a universal density approximator that can capture skewness and multimodality. However, this extension would break the reducibility properties of Gaussians on which Appendix A.1 relies. The reason our current objective (Equation (2)) can be efficiently optimized lies in closed-form solutions of certain Gaussian integrals, convolutions, and marginalizations, as well as differentiable probabilistic mappings. After introducing flow models:
>         - The kernel integrals involved in Equation (2) would no longer have closed-form expressions and would need to be approximated via two-dimensional Monte Carlo sampling (corresponding to the two random variables in pairwise preferences) to estimate expectations and gradients, which would significantly increase optimization variance and computational cost and place higher demands on convergence stability.
>         - Since the analytical form of the overall model would no longer maintain the structure of Equation (11), the deterministic mechanism of uncertainty emergence in Appendix A.2 (stemming from variance propagation and marginalization under Gaussian structure) would no longer be applicable. In other words, the uncertainty properties of the extended model would become strongly dependent on the specific flow mappings and numerical approximations, making it difficult to provide interpretability guarantees equivalent to those in the current manuscript.
>
> + W3: Thank you very much for the key suggestion. As a hyperparameter when applying PURM to RLHF, $\lambda$ indeed depends on a reasonable choice. According to the results in Appendix C.2, we find that $\lambda$ exhibits a pattern of first increasing and then decreasing with respect to RL performance. Therefore, we believe lambda can be selected based on the following strategy: first take $\lambda = 0$ as the baseline result (degenerating to BTRM), then take values at certain intervals thereafter (5, 10, 15, …), and when we observe a degradation in RL performance at some selection, perform a binary search. Thank you again for the suggestion; we will incorporate these insights into the ablation section in the revised version to help readers better utilize PURM.
>
> ### Summary:
> + Again, thank you for your thoughtful and profound suggestions; they greatly help improve the theoretical soundness, novelty, and completeness of this paper. We would be very happy to continue the discussion with you!

---

> > ### Author Response · Authors · 2025-11-15
> > **Reference**
> >
> > [1] Lou, Xingzhou, et al. "Uncertainty-aware reward model: Teaching reward models to know what is unknown." arXiv preprint arXiv:2410.00847 (2024).
> >
> > [2] Deep Reinforcement Learning from Human Preferences. (NeurIPS 2017)
> >
> > [3] Yuntao Bai, et al. "Training a Helpful and Harmless Assistant with RLHF.” arXiv preprint arXiv:2204.05862 (2022)
> >
> > [4] Tipping, M. E., & Bishop, C. M. (1999). Probabilistic principal component analysis. Journal of the Royal Statistical Society Series B: Statistical Methodology, 61(3), 611-622.
> >
> > [5] Auto-Encoding Variational Bayes. (ICLR 2014)

---

> ### Author Response · Authors · 2025-11-24
>
> Once again, we sincerely appreciate your review and the effort you devoted. We have uploaded the revised manuscript via the Rebuttal Revision and made corresponding corrections in Appendix C.2 and Appendix D based on your comments.

---

> ### Author Response · Authors · 2025-11-28
>
> Dear Reviewer CaMJ, thank you again for your time and effort in reviewing our manuscript. We hope our responses have adequately addressed your concerns. We are happy to engage in further discussion if you have any additional questions.

---

### Official Review · Reviewer_CysS · 2025-10-31

**Soundness:** 3
**Presentation:** 1
**Contribution:** 2
**Rating:** 2
**Confidence:** 3

**Summary:**

This paper proposes Probabilistic Uncertain Reward Model (PURM) as an extension of BTRM used in RLHF. Instead of modeling scalar rewards, PURM represents each reward as a Gaussian distribution parameterized by a mean $\mu$ and a standard deviation $\sigma$. This probabilistic formulation aims to capture uncertainty and mitigate reward hacking.

Parts of this review were discussed with a colleague to ensure clarity and accuracy.

**Contributions:**
1. Introduces a probabilistic variant of BTRM that outputs a Gaussian reward distribution rather than a scalar value.
2. Derives a tractable Monte Carlo–based training objective for learning from preference data.
3. Proposes a novel use of the Bhattacharyya Coefficient to quantify uncertainty in reward modeling.
4. Integrates uncertainty into RLHF by penalizing uncertain rewards to mitigate reward hacking.

**Strengths:**

1. The experimental results are strong;
2. The idea of modeling reward uncertainty in RLHF is interesting and conceptually aligns with the intuition that reward confidence should guide policy learning.

**Weaknesses:**

1. The method makes the reward modeling problem much more complicated than necessary. The probabilistic formulation and uncertainty estimation introduce large computational overhead. It would be helpful if the authors could justify whether such improvements are worth the added cost in real-world RLHF settings.
2. The derivation in Eq. (7)–(9) treats the pairwise reward difference $r_1 - r_2$ as Gaussian, but the validity of this assumption is not discussed. Since both $r_1$ and $r_2$ are modeled as independent Gaussian variables, this implicitly assumes independence between responses, which is unrealistic in preference data (they are conditioned on the same prompt $x$).
3. The paper approximates the intractable sigmoid–Gaussian integral using Monte Carlo sampling, but does not discuss the computational cost of this approximation during large-scale training.
4. The proposed use of the Bhattacharyya coefficient as a global uncertainty measure (Eq. 14–15) is questionable. Averaging pairwise overlaps with a random subset of the dataset (Eq. 16) is computationally heavy and not theoretically grounded as an uncertainty estimator.
typographical issues:
1. Inappropriate citation styles: all references are in `\citep` form. Mixing `\citet` and `\citep` properly would improve readability.
2. Differential notation `d` in `dz`, `dw` and `sigmoid` should be typeset as an operator (e.g.`\mathrm{d}z`, `\mathrm{d}w` and `\sigma(z)`).

**Questions:**

See weaknesses.

---

> ### Author Response · Authors · 2025-11-15
>
> ## Thank you very much for reviewing our paper. Below are our responses to your comments.
> ### Regarding strengths:
> + Thank you for recognizing the effectiveness of our experiments and the intuitive idea that uncertainty-guided RLHF alleviates reward hacking.
> ### Regarding weaknesses:
> + W1: This is an excellent question! We extended BTRM with a new capability dimension (i.e., uncertainty), and whether this introduces excessive additional computational cost is indeed worth examining. We conducted experiments; in line 262 and Appendix C.4, we present the training and inference computational overhead of PURM relative to other methods. As can be seen, PURM introduces almost no extra overhead compared to BTRM (because we minimized it via CUDA-based batched tensor operations during training and testing). Thank you for the suggestion; we will add this conclusion to the main text in the revised version to facilitate readers’ comparison of the computational cost of PURM and other methods.
> + W2: Thank you very much for the suggestion; this is a worthwhile topic to discuss. First, I guess you are referring to Equations (1)–(3)? Second, in deriving Equations (1)–(2), we indeed implicitly assumed the independence of $\mathcal{N}(r_1)$ and $\mathcal{N}(r_2)$, but this independence is conditioned on $x$, i.e., $r_1 \perp r_2 | x$ , given a prompt $x$, the quality scores of two responses should be independent of each other because the responses are produced by two different models/annotators. Consequently, $z = r1 − r2$ follows a new Gaussian distribution, which is the conclusion we derive; the detailed derivation is mentioned at line 147 and in Appendix A. Thank you again; we will clarify this point near Equations (1)–(3) in the revised version.
> + W3: Thank you for the suggestion. Our results in line 262 and Appendix C.4 show that this sampling incurs only a small additional computational cost (because we only need to sample n noise $\epsilon$ values from the standard normal distribution and then use the reparameterization trick $z = \mu + \sigma * \epsilon$ to obtain all z samples in one shot, which is very efficient). We apologize for overlooking this in the main text (only shown in line 820 in appendix); we will add the corresponding description in the revised version.
> + W4: This is also an excellent question! First, regarding the computational overhead of uncertainty based on the Bhattacharyya coefficient, we also show in line 262 and Appendix C.4 that it does not bring significant extra cost (because we can efficiently obtain the uncertainty u via CUDA-based batched tensor operations). Second, for the definition of uncertainty, prior work [1] also considers probabilistic modeling to be the most natural definition of reward uncertainty, and through the BT model we achieve the natural emergence of such a probability distribution via maximum likelihood from preference data. However, current RL requires a scalar reward rather than a reward distribution, so we convert this uncertainty into a scalar value that can be fused with the reward through an operation similar to marginalization (Equations (7), (9)). We specifically choose the Bhattacharyya coefficient because it is a good measure between distributions. Compared with directly taking sigma as the uncertainty (which ignores the effect of $\mu$) and taking $p(r_1 > r_2)$ as the pairwise degree of overlap (which would be fixed at 0.5 when $\mu_1 = \mu_2$, making $\sigma$ no longer matter), the Bhattacharyya coefficient effectively utilizes both mu and sigma to provide an effective uncertainty estimate, as reflected in Section 3.2 (Uncertainty Evaluation). Thank you again for the very careful and detailed question; it is very valuable for our work.
> Regarding typographical issues:
> + W5: Thank you very much for the suggestion. We will adjust citation commands according to the context in the revised version to achieve better presentation.
> + W6: We apologize for the inconvenience in reading. For differential notation, we will modify all instances in the revised version as you suggested; regarding the sigmoid function, we chose not to use the sigma symbol because our formulas contain many standard deviation variables ($\sigma_1$, $\sigma_2$), and we wanted to avoid confusion with them.
>
> ### Summary:
> Again, thank you for your thoughtful and profound suggestions; they greatly help improve the quality and readability of this paper. We would be very happy to continue the discussion with you!
>
>
> [1] Reinforcement Learning Under Uncertainty: Expected Versus Unexpected Uncertainty and State Versus Reward Uncertainty

---

> ### Author Response · Authors · 2025-11-24
>
> Once again, we sincerely appreciate your review and the time you invested! We have uploaded the revised manuscript via the Rebuttal Revision and made corresponding corrections throughout (including the explanation of conditional independence at line 143, as well as the mathematical notation and citations throughout the paper) based on your comments.

---

> ### Author Response · Authors · 2025-11-28
>
> Dear Reviewer CysS, thank you again for your time and effort in reviewing our manuscript. We hope our responses have adequately addressed your concerns. We are happy to engage in further discussion if you have any additional questions.

---

### Official Review · Reviewer_TxYA · 2025-11-03

**Soundness:** 2
**Presentation:** 3
**Contribution:** 3
**Rating:** 2
**Confidence:** 3

**Summary:**

This paper proposes the *Probabilistic Uncertain Reward Model (PURM)*, which extends the traditional Bradley–Terry Reward Model (BTRM) by introducing a probabilistic framework that models reward distributions rather than point estimates. PURM further quantifies uncertainty via the Bhattacharyya Coefficient, allowing uncertainty-aware reward penalties during RLHF to mitigate reward hacking. Empirical results show that PURM improves stability and achieves higher win rates over BTRM and other uncertain reward models.

**Strengths:**

1. The proposed method is simple, intuitive, and easy to implement. It appears effective in practice and introduces little additional latency compared to BTRM.
2. The *Uncertainty Evaluation* section presents particularly interesting observations, especially that PURM can adjust its reward uncertainty when training data labels are randomly flipped, while other baselines cannot.
3. The paper is well-written, with smooth logical flow and clear presentation of both the intuition and methodology.

**Weaknesses:**

1. **Conceptual novelty and related work.**
    The core idea, i.e. replacing a scalar reward with a Gaussian distribution and introducing an uncertainty-based penalty, is quite straightforward. I am surprised that such a distributional approach to reward modeling has not been explored before. I am not an expert in this subarea, but I found several potentially related works that are not discussed in the paper:

    - *Bayesian Reward Models for LLM Alignment*, ICML 2024 (Workshop)
    - *Active Preference-Based Gaussian Process Regression for Reward Learning*, RSS 2020
    - *Know What You Don’t Know: Uncertainty Calibration of Process Reward Models*, arXiv:2506.09338
    - *Aligning Crowd Feedback via Distributional Preference Reward Modelling*, ICLR 2025 (Workshop)

    If these works are relevant, the authors should position PURM more clearly relative to them, clarify its distinctive contributions, and include comparative experiments. If they are not directly related, it would still be valuable to explain *why* distributional reward modeling has received little prior attention.

2. **Empirical claims need stronger support.**

    - The choice of the Bhattacharyya Coefficient (BC) as the uncertainty measure is insufficiently justified. In Appendix C.1, its performance is not substantially better than simply using standard deviation, and this difference could likely be offset by tuning the hyperparameter λ.
    - Line 264 (“We attribute this to the fact that …”) asserts a causal interpretation that is not supported by explicit ablation or visualization.
    - Section 3.3 shows improved RLHF performance, but it does not clearly demonstrate that PURM mitigates *reward hacking* per se; it could simply reflect better learned reward.

3. **Limited exploration of downstream behavior.**
    The empirical analysis in Section 3.2 is incomplete. More fine-grained studies would greatly strengthen the paper, examples are:

    - Does PURM also improve policy model's robustness to noisy preference data or out-of-distribution (OOD) evaluation tasks?
    - How does it perform on tasks with inherently low reward noise, such as code or math reasoning where unit-test-based rewards are nearly deterministic?

    Exploring such aspects would definitely help the community understand the broader implications of probabilistic reward modeling.

------

### **Minor Issues**

1. Line 382: *“GPT-4o Hurst et al. (2024) is used …”* should read *“In Hurst et al. (2024), GPT-4o is used …”*.
2. The figures contain text that is too small to be readable after printing; font sizes should be increased for accessibility.

------

**Questions:**

See the concerns noted in the *Weaknesses* section. My current rating is deliberately conservative, but I would be happy to engage in discussion and to raise my ratings accordingly it if the authors address these issues.

---

> ### Author Response · Authors · 2025-11-15
>
> ## Thank you very much for reviewing this paper. Below are our responses to your comments.
>
> ### Regarding strengths
> + Thank you for recognizing the core motivation of this paper and the simplicity, intuitiveness, and effectiveness of the method.
> + We are likewise pleased that PURM exhibits corresponding uncertainty-awareness under two different paradigms of introducing uncertainty.
>
> ### Regarding weaknesses:
> + W1: Thank you very much for your suggestion! For the potentially related literature you provided, we clarify the differences in motivation and methodology between this paper and those works as follows:
>     - Literature [1]: Similar to literature [5] mentioned in this paper, it requires introducing human/model-labeled absolute reward values for maximum-likelihood training (line 454). Existing work [6, 7] generally considers absolute rewards to be often insufficiently robust, and that a better approach is to train reward models using human-labeled preference data. Therefore, unlike [1, 5], our method PURM does not require introducing human-labeled absolute reward values; instead, it can be trained directly on existing conventional preference data, allowing reward estimation and uncertainty estimation to spontaneously emerge in the process. The entire process has theoretical guarantees (Appendix A.1), can be implemented in a very lightweight manner (Appendix B), and runs with minimal additional computational cost (Appendix C.4). We consider this the core contribution of this paper.
>     - Literature [2]: Its starting point is not Bradley–Terry reward modeling, and because it studies time-sequential robotic scenarios, it defines the reward function as a Gaussian Process.
>     - Literature [3]: Its starting point is not Bradley–Terry reward modeling, and what it models is process reward (whereas this paper models outcome reward), primarily applying data collection and Quantile Regression methods.
>     - Literature [4]: It mainly characterizes the categorical distribution of the reward model across multiple preference categories (e.g., “Helpfulness,” “Harmlessness”), rather than the probability distribution of rewards in the real-number space. In summary, we believe that distributional reward models have received limited prior attention because their distributions are difficult to model. Unlike language models, which can learn a discrete distribution over a vocabulary via softmax, reward models usually need to perform scoring in real-number space. Therefore, an important contribution of this paper is that, by generalizing the BT model for preference learning, the reward model can spontaneously emerge with an estimate of the reward distribution. Thank you again for your valuable suggestions; we will supplement the revised version with the related literature above and the corresponding discussion and analysis to better present the motivation and contribution of this paper.

---

> > ### Author Response · Authors · 2025-11-15
> >
> > + W2:
> >      - Thank you very much for your suggestion. We have begun performing multi-random-seed repetitions of the core experiments to improve the credibility of the statistical results, and we will tune lambda according to your suggestion to obtain its optimal performance. Since reinforcement learning experiments have high computational overhead, these experiments will take some time; we will complete them as soon as possible and submit updated results at a later date. Thank you again for your constructive feedback.
> >     - This is a very good question! In lines 264–268 we indeed only provided some explanation for the experimental results based on the analysis in Appendix A.2. To further provide evidence for this inference, we designed the following experiment: we still perform forward with the current PURM model, but when computing the loss we only select the left-side mu head and compute according to the BT loss. We name this PURM-degenerate version, and we test it on RewardBench as well.
> > As shown in the Table below, we can see that on RewardBench, PURM-degrade’s performance has essentially reverted to that of BTRM, which confirms that PURM’s gains on RewardBench are not due to a simple architectural change, but rather stem from the combination of architectural changes and maximum-likelihood training based on the reward distribution.
> >
> >     - This is a very interesting question! We can discuss it from the following two aspects:
> >         - a) First, we believe that training a better reward model alleviates reward hacking, because the more faithful our reward model is and the closer it is to human preferences, the more general the policy model’s learned policy will be, and it will not overfit the training prompts; a perfect reward function (usually rule-based, e.g., the win/loss signal in Go, the correctness signal in math problems) would not suffer from reward hacking [8]. For the non-verifiable tasks we discuss, since there is no rule-based reward, we must use a NN-based reward model to approximate the true reward (i.e., human preference). Therefore, we believe that in this scenario, alleviating reward hacking is essentially equivalent to improving the generalization ability of the NN-based reward model.
> >        - b) If you suspect that the gain in RL comes from better reward estimation rather than uncertainty estimation, then our ablation experiment in Figure 5(b) can demonstrate this well: when $\lambda = 0$, our PURM degenerates into an ordinary reward model that does not use uncertainty, and the RL performance likewise degenerates to be comparable to BTRM; when we introduce uncertainty-based reward shaping, RL performance improves significantly.
> >
> > | Domain | Metric | BTRM | PURM | PURM-degrade |
> > |--------|--------|------|------|--------------|
> > | Chat | ACC ↑ | 94.69 | 96.37 |91.34|
> > | Chat | NLL ↓ | 0.179 | 0.151 |0.253|
> > | Chat Hard | ACC ↑ | 48.79 | 50.22 |47.92|
> > | Chat Hard | NLL ↓ | 1.040 | 1.020 |1.047|
> > | Safety | ACC ↑ | 75.29 | 76.82 |77.88|
> > | Safety | NLL ↓ | 0.519 | 0.507 |0.534|
> > | Reasoning | ACC ↑ | 93.98 | 96.29 |93.94|
> > | Reasoning | NLL ↓ | 0.382 | 0.354 |0.437|
> > | Overall | ACC ↑ | 82.90 | 84.20 |82.40|
> > | Overall | NLL ↓ | 0.492 | 0.468 |0.532|

---

> > > ### Author Response · Authors · 2025-11-15
> > >
> > > - W3:
> > >     - You have proposed a very good suggestion! This can indeed effectively enhance the validity of our evaluation of PURM uncertainty in Section 3.2. According to your suggestion, we are conducting the following experiments: a) Train BTRM and PURM respectively using data with 20% flipped noise, and perform RL training based on these two reward models. b) Train BTRM and PURM respectively using only reasoning data, and perform RL training based on these two reward models on HH-RLHF (primarily helpfulness and harmlessness) data. Since reinforcement learning experiments have high computational overhead, running these experiments will take some time; we will complete them as soon as possible and submit the experimental results at a later date. Thank you again for your constructive feedback.
> > >     - Because in code and math scenarios there are usually verifiable rewards (lines 27–29), these scenarios generally only require rule-based reward to perform RLVR and can achieve very good results, and they do not have the reward hacking risk that reward models do [8]. Therefore, this work mainly discusses the capability of reward models in non-verifiable tasks (line 30).
> > >
> > > ### Regarding Minor Issues
> > > + Thank you very much for your suggestion. We indeed used the \citep command throughout the text without considering the grammar of the surrounding context. We will carefully fix this in the revised version to provide a better reading experience for readers.
> > > + Thank you very much for your suggestion as well. We will increase the font size in the figures in the revised version.
> > >
> > > ### Summary
> > > + Thank you again for your thoughtful and insightful suggestions. They are of great help in improving the quality and logic of this paper, and we are very willing to continue the discussion with you!
> > >
> > >
> > >
> > >
> > > [1] Bayesian Reward Models for LLM Alignment, ICML 2024 (Workshop)
> > >
> > > [2] Active Preference-Based Gaussian Process Regression for Reward Learning, RSS 2020
> > >
> > > [3] Know What You Don’t Know: Uncertainty Calibration of Process Reward Models, arXiv:2506.09338
> > >
> > > [4] Aligning Crowd Feedback via Distributional Preference Reward Modelling, ICLR 2025 (Workshop)
> > >
> > > [5] Uncertainty-aware reward model: Teaching reward models to know what is unknown. arXiv preprint arXiv:2410.00847 (2024).
> > >
> > > [6] Deep Reinforcement Learning from Human Preferences. (NeurIPS 2017)
> > >
> > > [7] Training a Helpful and Harmless Assistant with RLHF. arXiv preprint arXiv:2204.05862 (2022)
> > >
> > > [8] DeepSeek-R1: Incentivizing Reasoning Capability in LLMs via Reinforcement Learning arXiv preprint arXiv:2501.12948 (2025)

---

> > > ### Author Response · Authors · 2025-11-30
> > >
> > > Additionally, the RL training performance of PURM across multiple random seeds is now presented in Figure 8 of the revised manuscript. We appreciate your suggestion.

---

### Comment · Reviewer_TxYA · 2025-11-22
**Replying to the authors**

Thank you for the prompt and detailed response. Your clarifications for **W1** and **W2.2** have largely addressed my concerns, and the planned experiments for **W2.1** and **W3.1** also appear reasonable. Below are my remaining comments.

**(W2.3) On the claim of mitigating *reward hacking***

In the RLHF literature, *reward hacking* usually refers to cases where a reward model with poor generalization assigns abnormally high rewards to highly atypical out-of-distribution responses or pathological token patterns (e.g., excessive repetition, unnatural multilingual mixtures, etc. [1]). During RL, the policy model then exploits such patterns, leading to artificially inflated rewards while actual performance deteriorates sharply — for instance, producing hallucinations or code that cannot compile [2].

By contrast, improving the accuracy of the reward model *within* the training distribution (e.g., better separation between positive and negative samples) can also enhance policy performance, but this does not necessarily imply a genuine reduction in reward hacking. Although the two are correlated, they are conceptually not equivalent.

Therefore, to substantiate the claim that PURM *mitigates reward hacking*, I strongly encourage the authors to include more targeted and convincing evidence. For example, it would be helpful to report: What proportion of policy errors under BTRM can be explicitly attributed to recognizable reward-hacking behaviors? After switching to PURM, how significantly are such errors reduced? Such analyses would greatly strengthen the causal interpretation behind the claim.

**2. (W3.2) On finer-grained empirical analysis and scope of applicability**

I fully agree that scenarios with verifiable rewards are not the focus of this paper. My intention in raising this point was to encourage a more fine-grained breakdown of the empirical results , especially in **Section 3.2 (reward evaluation)** and **Section 3.3 (downstream performance)**.

For instance:

- On which types of tasks does PURM yield the largest improvements? What might be the underlying reasons?
- Is there a monotonic relationship between reward noise and the gain from PURM? For example, does PURM provide only marginal benefits in low-noise domains, but substantial improvements in highly noisy ones (hence my reference to code and math tasks)?

As stated earlier, these are illustrative directions rather than mandatory questions (and I would be please to see the authors explore beyond my requests). The core motivation is that PURM represents a *paradigm-level* modification. Therefore, I would expect additional empirical evidence that clarifies: its scope of effectiveness, and the insights that can be distilled for the broader community.

Given the relative simplicity of the method, such analyses would significantly enhance the overall impact and interpretability of the paper.

------

**Additional remarks**

I have also reviewed other reviewers’ comments and currently have no further concerns. Gaussian assumption is  acceptable to me if the experimental results are strong, as the Bradley–Terry model itself is already a strong behavioral assumption. I look forward to the new experimental results and the revised version of your paper.

------

**References**

[1] *Uncovering the Impact of Chain-of-Thought Reasoning for Direct Preference Optimization: Lessons from Text-to-SQL*, ACL 2025

[2] *Is DPO Superior to PPO for LLM Alignment? A Comprehensive Study*, ICML 2025

---

> ### Author Response · Authors · 2025-11-30
>
> ### Response to Reviewer
>
> We sincerely appreciate your constructive feedback. Your suggestions regarding the distinction between general performance improvements and reward hacking mitigation (W2.3), as well as the need for a finer-grained empirical analysis of the method's scope (W3.2), have significantly strengthened our paper.
>
> In response to your comments, we have added two new subsections in Appendix C: "PURM vs BTRM in Noisy and OOD settings" and "Hacking Analysis". Below, we detail how these additional experiments address your concerns.
>
> ### (W3.2) On finer-grained empirical analysis and scope of applicability
>
> We fully agree with your insight that exploring the relationship between reward noise and PURM’s gains is crucial for understanding the method's scope. As suggested, we conducted additional experiments to compare PURM and BTRM under different noise levels (0.2 vs. 0.4) and in Out-of-Distribution (OOD) settings.
>
> New Experimental Results: As illustrated in the newly added Figure 9 (included in the revised manuscript), our findings confirm a monotonic relationship between data noise/uncertainty and the benefits of PURM:
>
> Low Noise (0.2): PURM maintains a slight advantage over BTRM.
> High Noise (0.4): PURM shows a pronounced performance gap over BTRM.
> OOD Settings: PURM achieves the most significant gain (over 5% win rate improvement).
> These results empirically define the scope of PURM: it is particularly effective in high-noise environments and OOD scenarios where epistemic uncertainty is high, validating that PURM effectively leverages uncertainty estimation to improve robustness.
>
> ### (W2.3) On the claim of mitigating reward hacking
>
> Thank you for pushing us to substantiate the claim regarding reward hacking. We agree that simply improving general accuracy is not equivalent to mitigating hacking. To address this, we conducted a targeted analysis on the HH-RLHF (Safety and Helpfulness, 8,552 samples).
>
> In the context of Safety RLHF, a common form of "hacking" or "policy exploitation" occurs when the policy exploits the Reward Model's bias towards safety by over-refusing benign queries. The policy learns that producing "refusal tokens" is a shortcut to maximizing rewards, even when the prompt is safe. This leads to a sharp deterioration in actual performance (Helpfulness).
>
> New Hacking Analysis: We measured the specific "policy errors" where the model refuses to answer safe prompts. As shown in the newly added Table 3:
>
> Quantifying the Exploitation (BTRM): BTRM exhibits severe over-defensive behavior with a Precision of only 0.4937. This implies that more than half of its refusals are False Positives—it is "hacking" the safety objective by refusing safe queries (~16% of them) to maximize the reward.
> Mitigation by PURM: By incorporating uncertainty, PURM effectively penalizes this blind "refusal shortcut." PURM improves Precision to 0.6794 and Safe QA Accuracy (Helpfulness) to 93.60%.
> This evidence supports the causal interpretation: PURM prevents the policy from exploiting the "refusal bias" in the reward model, resulting in a policy that answers significantly more safe questions correctly while maintaining robust safety recall. This demonstrates a genuine reduction in the policy's tendency to exploit pathological patterns (unnecessary refusals) for reward gain.

---

### Comment · Area_Chair_Rt3E · 2025-11-26

Dear all colleagues,

Thank you all for your hard work on the reviews. Now, we will proceed to the stage of discussion with the authors. The authors have provided their rebuttal and revision. Can you review it and confirm if there are any further concerns about the work?

Regards,

AC

---

### Meta-Review · Area_Chair_gzUq · 2026-01-02

**Summary:**

This paper introduces the Probabilistic Uncertain Reward Model (PURM), which serves as an advanced extension of the traditional Bradley–Terry reward model. Unlike the classic model, which generates a fixed scalar reward, the PURM model provides a more nuanced output in the form of parameters representing a Gaussian distribution, specifically the mean (μ) and standard deviation (σ). By doing so, PURM captures the inherent uncertainty associated with the reward signals, allowing for a probabilistic interpretation of the rewards rather than a deterministic one. To quantify the degree of overlap between different reward distributions, the authors incorporate the Bhattacharyya coefficient. This coefficient acts as a measure of similarity between the distributions, thus facilitating an effective mechanism for uncertainty quantification. The authors assert that this innovative approach enhances the stability of Reinforcement Learning from Human Feedback (RLHF) training processes. By accounting for uncertainties in the reward signals, PURM aims to reduce the risks of reward hacking, where agents exploit weaknesses in the reward structure to achieve undesired behaviors. Overall, the proposed model represents a significant advancement in the field of reinforcement learning by integrating probabilistic elements to better manage uncertainty and improve training robustness.

**Reviewer Concerns:**

The reviewers TxYA and CaMJ believe the novelty of the paper is limited, since there are several existing work on Distributional reward modeling and uncertainty quantification, for example, URM.
The experiments of the paper are not quite strong; two reviewers raised significant concerns on the experiment settings and results. The authors have added many experiments, the reviewers are not convinced by the results.
There are also several concerns like paper assumption, hyperparameter, evaluation metric and among others.

**Reviewer Scores:**

The initial scores are 2,2,4,4. After rebuttal, one reviewer with 4 has increased to 6. I believe, by the authors' effort, the first 2 could be turned to 4. However, the final cores of 4,2,4,6 are still not enough to achieve the boardline bar of ICLR.

---

### Decision · Program_Chairs · 2026-01-26

Reject